# A Brief Review on Selected Applications of Hybrid Materials Based on Functionalized Cage-like Silsesquioxanes

**DOI:** 10.3390/polym15061452

**Published:** 2023-03-14

**Authors:** Łukasz John, Jolanta Ejfler

**Affiliations:** Faculty of Chemistry, University of Wrocław, 14 F. Joliot-Curie, 50-383 Wrocław, Poland

**Keywords:** polyhedral oligomeric silsesquioxanes (POSS), double-decker silsesquioxanes (DDSQ), biomaterials engineering, flame retardants, Ziegler-Natta heterogenous catalysis

## Abstract

Rapid developments in materials engineering are accompanied by the equally rapid development of new technologies, which are now increasingly used in various branches of our life. The current research trend concerns the development of methods for obtaining new materials engineering systems and searching for relationships between the structure and physicochemical properties. A recent increase in the demand for well-defined and thermally stable systems has highlighted the importance of polyhedral oligomeric silsesquioxane (POSS) and double-decker silsesquioxane (DDSQ) architectures. This short review focuses on these two groups of silsesquioxane-based materials and their selected applications. This fascinating field of hybrid species has attracted considerable attention due to their daily applications with unique capabilities and their great potential, among others, in biomaterials as components of hydrogel networks, components in biofabrication techniques, and promising building blocks of DDSQ-based biohybrids. Moreover, they constitute attractive systems applied in materials engineering, including flame retardant nanocomposites and components of the heterogeneous Ziegler-Natta-type catalytic system.

## 1. Introduction

The public has dealt with many modern technology-related topics in the last decade. This phenomenon is directly allied with rapidly developing materials engineering. Whereas a few decades ago, the terms “self-assembling systems”, “self-healing materials”, or “artificial intelligence” sounded like a distant future, now it seems that these concepts are not detached from real applications. The rapidly developing materials engineering is accompanied by the equally rapid development of new technologies, which are now increasingly used in various branches of our life.

This short review focuses on selected silicon-based materials and their applications. Among these species, we will focus on two groups of cage-like silsesquioxanes, polyhedral oligomeric silsesquioxanes (POSSs) and double-decker silsesquioxanes (DDSQs). The current research trend concerns the development of methods for obtaining new materials for materials engineering and the search for relationships between the structure and physicochemical and mechanical properties. This fascinating field of hybrid species has attracted considerable attention due to their daily applications with unique capabilities and their great potential, among others, in biomaterials and materials engineering, including flame retardant nanocomposites and catalysis.

## 2. What Is Unique about Silicon-Based Species?

Silicon is one of the most common elements in the earth’s crust due to its abundance in various minerals, e.g., aluminosilicates. As an element of the fourteenth group of the periodic table, it has many similarities to carbon compounds. Comparing the silicon–carbon and carbon–carbon bonds, it can be seen that the first one is longer and weaker. Si–C bond length is 1.87 Å, with a bond energy of 337 kJ/mol, whereas C–C bond length equals 1.53 Å, manifesting a binding energy of 371 kJ/mol. The electronegativity of the carbon atom is greater than that of the silicon atom. Therefore, the Si–C bond is polarized towards the carbon atom and can be broken more easily than typical C–C bonds. A much stronger and longer bond concerning C–O (345 kJ/mol, 1.41 Å) is the Si–O bond (426 kJ/mol, 1.64 Å). The significant difference in the electronegativity of silicon and oxygen, which equal 1.8 and 3.5, respectively, makes the siloxane bonds ionic at 51%, which means their high susceptibility to heterolytic cleavage and high thermal stability. The longer length and half-ionic character of the Si–O bond contribute to much larger angles between the Si–O–Si bonds (143°) than C–O–C (112°) in the ethers, lowering the rotational barrier around the Si–O bond (2.5 kJ/mol) and the energy barrier of linearization of the Si–O–Si angle (1.3 kJ/mol), which consequently affects the extraordinary flexibility of the siloxane chain. The Si–O bond is one of silicon’s most thermally stable bonds, with 110 kcal/mol dissociation energy [1,2]. Large alkyl substituents or aryl groups reduce the freedom of rotation around the Si–O–Si bond, causing an increase in the glass transition temperature. Compared to C–H, the reactivity of the Si–H bond is also worth noting. The C–H bond is stronger (413 kJ/mol) and inert in general, whereas Si–H (average bond energy equals 393 kJ/mol) is liable to hydrolysis, which is crucial in hydrosilylation reactions [3].

Thermodynamically stable bonds typical of silicon compounds include siloxane bridges –Si–O–Si–, while silicon bonds –Si–Si–Si– do not form stable moieties. In the case of carbon compounds in the alkane series, the situation is reversed because the carbon chains –C–C–C– are thermodynamically stable, and the –C–O–C– type connections in most cases show less stability (here, the exceptions include, e.g., diethyl ether). It is similar in the case of cyclic forms, where the thermodynamically stable ones are those with Si–O building units and carbon rings, respectively; an example of a stable ring with an –C–O–C– linkage is tetrahydrofuran. In silicon compounds, polarized covalent bonds are preferred, while in carbon compounds, there are covalent, slightly polarized sigma bonds resulting from the imposition of appropriate atomic orbitals. In silicon compounds, silicon’s unfilled 3*d* atomic orbital participates in the Si–O bond formation. On the other hand, molecular orbitals of the *π* type also form multiple bonds between carbon atoms, resulting from overlapping atomic *p* orbitals. In turn, the similarities of carbon and silicon compounds include: (i) the formation of a wide range of connections related to straight, branched, or ring sequences –C–C–C– and –Si–O–Si–, (ii) tetrahedral geometry of silicon and carbon atoms, (iii) the ability to form oligo- and polymeric systems, (iv) undergoing spontaneous transformations in changing external conditions, and (v) natural occurrence in nature numerous compounds of these elements [4].

All those mentioned-above facts make silicon-based compounds unusual and determine their unique architecture, thermal stability, and reactivity.

## 3. Cage-like Silsesquioxanes

The first oligomeric organosilsesquioxane was discovered and isolated by Scott in 1946 due to the thermolysis of products formed during the hydrolysis of MeSiCl_3_ and Me_2_SiCl_2_ [5]. Since then, an avalanche of research on this group of organosilicons has begun, including the cage structures. Silsesquioxane compounds belong to organosilicon species with the general formula [RSiO_3/2_]*_n_* (R = H, alkyl, aryl, alkoxo; *n* = 4, 6, 8, 10, 12, 14, 16, 18), and they form various architectures (Figure 1) [6]. Hence, they can adopt cage-like or polymeric architectures with repeated Si–O–Si siloxane moieties and tetrahedral silicon vertices. In general, they are usually obtained in the hydrolysis of trifunctional organosilanes. In the literature, two classes of cage-like silsesquioxanes are known, polyhedral oligomeric silsesquioxanes (POSSs) and double-decker silsesquioxanes (DDSQs), which are briefly discussed below.

### 3.1. Polyhedral Oligomeric Silsesquioxanes (POSSs)

Polyhedral oligomeric silsesquioxanes constitute well–defined organosilicon compounds with a closed cage structure labeled T_6_, T_8_, T_10_, and T_12_ tetrahedral vertex, possessing 6, 8, 10, and 12 silicon atoms, respectively. These compounds result from hydrolysis and subsequent condensation of trifunctional RSiX_3_ silanes (where: X = Cl, OMe, OEt, etc.; R = *^n^*PrNH_2_, *^n^*PrOH, *^n^*PrSH, *^n^*PrN_3_, etc.) in the presence of acidic or basic catalysts. While the alkyl group forming the C–Si bond is inert in the hydrolysis reaction, the silyl group is labile and forms the Si–O–Si siloxane network. These silanes are readily hydrolyzed to form Si–OH silanol groups, which in further stages condense to form a siloxane cage. Among this group of cage-like silsesquioxanes, the most explored are T_8_ cage octasilsesquioxanes (Figure 2), which are highly soluble in organic solvents.

Various organic side groups can be attached to these compounds, which determine the structural and physicochemical properties of the resulting hybrid materials. Covalently bonded to the silicon core, the side arms are labile groups that enable various chemical reactions to be carried out on them and are effective nano-building blocks for, among others, supramolecular structures [7,8,9]. Depending on the number of reactive organic arms attached to the T_8_-type cage, there are mono-, di-, including *ortho*, *meta*, *para* diadducts, and octa-substituted derivatives (Figure 2) [10,11,12,13,14,15,16,17,18].

### 3.2. Double–Decker Silsesquioxanes (DDSQs)

Yoshida et al. reported double-decker silsesquioxanes for the first time in 2003 [19]. Since then, numerous papers have been published showing interesting chemistry and vast applications of these organosilicon compounds [20]. DDSQ comprises two cyclotetrasiloxane rings bound together by two oxygen atoms, with eight phenyl groups attached to silicon atoms. Additionally, two or four lateral silicon atoms can be attached to this inorganic core, as shown in Figure 3.

As can be seen in Figure 3, DDSQs form two kinds of architectures, namely closed with one (Figure 3I) or two (Figure 3II) reactive side arms attached to two lateral silicon atoms and open structures with one labile organic group bonded to every lateral Si atom (Figure 3III). DDSQs of I type are routinely obtained as the mixture of *cis* and *trans* geometrical isomers, corresponding to the spatial arrangements of the reactive and inert organic groups attached to the double-decker core. Such a mixture can effectively be separated by fractional crystallization [21].

## 4. Selected Biomaterials Based on Cage-like Silsesquioxanes

Exploring the research area related to broadly understood regenerative medicine and materials engineering is currently a fascinating and timely topic undertaken by many multidisciplinary scientific groups worldwide. At the end of the 1960s, research in biomaterials was initiated by the late Larry Hench and his co-workers, focusing on bioglasses based on oxide components. This tremendous breakthrough has led to commercial applications of the Na_2_O-CaO-P_2_O_5_-SiO_2_ system for bone repair, enamel-healing toothpaste, dental restoration, drug delivery, and so forth [22,23,24]. Then, the interests of scientists evolved toward polymeric materials, ceramics, and composites [25,26,27,28]. Currently, most attention is paid to mixed materials, often referred to as hybrid materials, mainly based on combinations of bioglasses and ceramics with appropriately modified polymer matrices [29,30] or hybrids in which a covalent bond connects individual components; for instance, polyhedral oligomeric silsesquioxanes (POSSs) and double-decker silsesquioxanes (DDSQs), in which the inorganic silsesquioxane core is bonded with organic side arms [7,20,31,32]. Multifunctional materials designed in this way fit into the current definition of third-generation biomaterials, the purpose of which is not only to replace damaged tissue fragments, but also to resorb them in the body and stimulate the host tissue to rebuild.

Organic-inorganic hybrid materials have recently gained the most interest among the materials used for implants, coatings, and fillings. Due to their high biocompatibility and ability to create chemical connections with living tissues, these biomaterials are considered the most valuable materials for biomedical applications. This group of hybrids includes caged silsesquioxanes (both POSSs and DDSQs), which combine the advantages of an inorganic core and modifiable organic groups anchored to it.

### 4.1. Cage-like Silsesquioxanes Incorporated into the Hydrogel Network

In recent years, hydrogel-based composites have been tested in various biomedical applications, such as medical dressing, wound-healing coatings, bone repair, tissue regeneration, drug delivery, and so forth. Increased interest in this type of materials derives from hydrogels’ good biocompatibility and biodegradability. However, their application is limited due to poor mechanical properties, low thermal stability, and rapid degradability in the organism. From this point of view, using cage-like silsesquioxanes as a sophisticated gel modifier, including cross-linking agent, reinforcement factor, building blocks for creating porous architectures, hydrophobic agent, etc., seems justified [33,34]. The resulting biomaterials may affect angiogenesis in bone defect repair, improve the adhesion of stem cells and proteins, and release growth factors, thereby promoting vascularization and hard tissue regeneration [35]. For instance, Z. Li et al. [36] studied the influence of cage-like silsesquioxanes on gelatin-based composite features using 3-glycidyloxypropyl-POSS (G-POSS) with various epoxy groups. In the next step, the resulting G-POSS was modified by polyetheramine (PEA) to achieve better solubility in aqueous media. Moreover, functionalization by PEA resulted in improved flexibility and thermal stability of the final P-G-POSS composite. In the next step, the hydrophilic composite was covalently incorporated into the gelatin network, owing to the reaction between amino and epoxy substituents. The results showed that the mechanical properties and thermal stability of the P-G-POSS/gelatin composite increased with an increasing amount of P-G-POSS in the gelatin matrix. This opens a new idea for modifying gel structures using POSS in water, which can be beneficial from a medial applications point of view.

High-strength hydrogels can also be obtained via the in-situ deposition of nanohydroxyapatite (HAP) after photopolymerization of gelatin methacryolyl (GelMA), quaternized chitosan (QCS) and modified nanoparticles of polyhedral oligomeric silsesquioxanes (POSS-Ac) [37]. Modified POSS was obtained in the substitution reaction of functionalized POSS with tertiary amine and hydroxyl groups (POSS-(OH)_32_) and acryloyl chloride (Figure 1). The primary aim of cage-like silsesquioxanes’ modifications was to increase dispersion while lowering the agglomeration of POSS nanoparticles in the polymer matrix. As a result, the final POSS-Ac structure allows it to play the role of inorganic crosslinkers, which can be successfully spread into an organic network. The resulting GelMA/QCS/POSS-Ac hydrogel manifested a highly interconnected matrix architecture and promising hydroxyapatite mineralization, much better than comparable hydrogels. Moreover, a complex hydrogel greatly facilitates the adhesion, viability, proliferation, and spreading of MC3T3-E1 cells and can promote the in-situ regeneration of bone defects.

Another example of the mechanically robust hydrogels facilitating hard tissue remodeling and regeneration through epigenetic modulation is gel based on cage-like silsesquioxane designated as an inorganic core surrounded by six disulfide-linked PEG shells and two 2-ureido-4[1H]-pyrimidinone (UPy) substituents (POSS-P_6_-U_2_) [38]. The resulting difunctionalized dual hybrid system promotes stem cells’ osteogenic capacity due to tunable structure and mechanical properties. The thiol-disulfide exchange reaction created a “living” network due to its pH-responsive “on/off” mode. At the same time, the UPy fragment reinforces the local microstructure responsible for mechanical properties. The entire multifunctional hybrid system possesses perfect cytocompatibility and bioactivity, making it attractive bedding for cell attachment, growth, and proliferation in vitro and in vivo.

In turn, B. Yu, W. Li, and J. Liao et al. reported on similar dual cross-linked functional hydrogel composed of polyethylene glycol diacrylate (PEGDA), thiolated chitosan doped with Ag nanoparticles (TCS@Ag), and POSS modified by dopamine and acryloyl chloride which constitutes the matrix for TCS@Ag [39]. The resulting PEGDA/TCS@Ag/POSS hydrogel was tested as a wound-healing agent. Furthermore, in vitro and in vivo experiments on antibacterial gel proved favorable to skin repair and regeneration capacity. It is also worthy that, due to convenient direct gelation, the functionalized POSS-reinforced hybrid hydrogel is a perfect candidate for surgical operation and has excellent therapeutic potential for cutaneous skin wound healing. Furthermore, the example of this composite showed that adding Ag^+^ ions to the hydrogel positively affects its antibacterial properties. Other frequently used admixtures in hybrid biomaterials include strontium and magnesium ions, which have, among others, potential anti-osteoporotic, anti-inflammatory, and bacteriostatic properties [29,40]. In this area, W. Li and W. Chen et al. reported on gelatin/strontium hydrogel containing dopamine-modified POSS (Dopa-POSS) [41]. Strontium ions spread in the arginine-glycine-aspartate matrix promote the proliferation and activity of the endothelial progenitor cells (EPCs). Moreover, the authors claim that the resulting Gel/Sr^2+^@POSS/EPCs hybrid composite accelerates vascularization, re-epithelialization with almost closed wounds, and collagen deposition, creating beneficial therapeutic bedding for burn wound healing. The dual-crosslinked hydrogel can also be considered a perfect matrix for magnesium cations. It was reported that the combination of gelatin methacryloyl (GelMA), thiolated chitosan (TCS), and functionalized POSS seems an attractive network for Mg-S bonds that fulfills the requirements for angio- and osteogenesis [42].

### 4.2. Cage-like Silsesquioxanes as Components in Biofabrication Techniques

Recently, an exciting and intensively explored area of research in advanced biomaterials is prototyping/fabrication systems [43]. The advantages of biofabrication are complete control of the geometry and porosity of materials and no need to use additional porogen to obtain a scaffold. Furthermore, various types of precursors, such as polymers, composites, hybrids, ceramics, bioglasses, and metals, can be used in prototyping methods. However, biofabrication techniques also have some disadvantages, including time-consuming procedures, high cost of equipment, temperature limitations, etc.

Three-dimensional biofabrication can be used to obtain hybrid scaffolds for supporting articular cartilage regeneration. As inks, biopolymers are frequently used. However, bio-inks based on polymers must be improved to achieve appropriate mechanical properties through physical interaction and chemical cross-linking. For instance, the composition of poly(lactic-co-glycolic acid) (PLGA)/polycaprolactone (PCL) can be supported by adding 3–5% (*w/v*) of POSS nanoparticles, which was reported by H. Mahdavi et al. [44]. The addition of cage-like silsesquioxane was also reinforced by the cartilage-derived extracellular matrix (ECM) particles to increase the hydrophilicity of the entire complex system. The authors showed that the two systems fulfill the best requirements of printed articular cartilage scaffold. First, the combination of PLGA/PCL/POSS (3%, *w/v*)/ECM (20%, *w/v*) manifested the best compressive modulus, whereas PLGA/PCL/POSS (5%, *w/v*)/ECM (20%, *w/v*) demonstrated high cell viability and biocompatibility. The resulting bio-ink compositions constitute an attractive scaffold for long-term regeneration during the healing process of cartilage tissue.

Another example of biofabrication with the participation of cage-like silsesquioxanes is the construction of biomimetic artificial intervertebral discs using electrospinning and three-dimensional printing. L. Lu et al. reported an artificial composite scaffold modified by PLLA octa-substituted cage-like silsesquioxane particles spread in a poly-L-lactide matrix (PLLA/POSS-(PLLA)_8_) [45]. The resulting biomaterial has been used to simulate the annulus fibrosus (AF). PLLA/POSS-(PLLA)_8_ component constitutes a part of the more complex system, including hydrogel (mixture of poly(ethylene glycol diacrylate) and gellan gum) loaded with bone marrow mesenchymal stem cells. POSS particles inside the composite significantly enhance the biocompatibility of poly-L-lactide, but primarily its strength and toughness. The architecture and mechanical properties of the resulting biomaterial are similar to those observed for natural intervertebral disc-supporting tissue regeneration and remodeling (Figure 4).

POSS nanoparticles can also be added to inks based on polycaprolactone (PCL)/poly(lactic-co-glycolic acid) (PLGA) to achieve better mechanical consistency and optimize the fabrication process [44].

Cage-like silsesquioxanes can also be the carrier for photoinitiators such as benzophenone derivatives [46]. The advantages of this type of system are lower cytotoxicity corresponding to other benzophenone species, low tendency to migrate, increased photoactivity, and many others. Furthermore, their use in three-dimensional printing creates high-resolution and accurate architectures. Moreover, methacrylphenyl POSS (MP-POSS) modified ultraviolet curing systems can be applied in 3D printing. Y. Cui et al. reported that the resulting MP-POSS possesses excellent compatibility and dispersibility under the action of N-vinylpyrrolidone dispersant [47]. As a photoinitiator, MS-POSS can form cross-link architecture with a concentration lower than 3% (wt). Moreover, adding modified POSS to the epoxy acrylate matrix significantly improved its mechanical properties and the polymer network’s tensile and impact fracture morphology.

In various orthopedic applications, polyetheretherketone (PEEK) implants are commonly used. However, the main disadvantages of using PEEK as an implant are the high cost of fabrication, lack of thermoformability, poor osseointegration, and biologically inert surface. Nevertheless, the PEEK-based scaffolds can be modified to avoid some drawbacks related to PEEK. For instance, D. Sun et al. described the bioprinting macroporous PEEK implant combined with methacrylated chitosan and polyhedral oligomeric silsesquioxane (CSMA/POSS) [48]. The resulting microporous scaffold was obtained through three-dimensional printing, sulfonation, and ultraviolet-induced graft polymerization (Figure 5). The biomineralization studies revealed that the obtained composite promotes protein adsorption and hydroxyapatite formation of the implant surface. Furthermore, in vitro and in vivo studies confirmed that a 3D-printed microporous PEEK-based scaffold provided a perfect environment for cell adhesion and proliferation, improving the osteogenic differentiation of rat bone marrow mesenchymal stem cells and encouraging osteogenesis in vivo compared to the pure PEEK implant.

It was also reported that the ink based on nacre, polyurethane (PU), and POSS nanoparticles (NPP composite) could be applied in the 3D printing of photo-crosslinked, anatomically tooth-shaped implants for alveolar ridge preservation after tooth extraction [49].

### 4.3. Double-Decker Silsesquioxanes as Novel and Promising Components of Hybrid Biomaterials

The skin is a vital barrier that protects the body from external threats, and its integrity is critical for overall health and well-being. However, exposure to various physical and environmental hazards makes the skin susceptible to injuries and wounds that can lead to impaired function and long-term consequences. To address this challenge, there is a constant need for innovative and effective wound-healing materials that can help to promote rapid and efficient tissue regeneration. The use of biomaterials in wound healing has undergone significant advancement in recent years, with the development of materials that range from simple cotton wool dressings to advanced skin substitutes that contain growth factors and cells that promote tissue regeneration. One class of biomaterials that has gained attention in this field is polyhedral oligomeric silsesquioxane-based (POSS-based) materials. These materials have demonstrated remarkable versatility and can be used for various applications, including tissue regeneration and wound healing [50]. However, despite their potential, functionalized POSS-based materials have been met with some challenges that need to be addressed. One of the main challenges in using POSS-based materials is the reactivity of silanol moieties in POSS cages with surfaces and their tendency to condense with other functional groups. This reactivity can block the functional groups responsible for cellular activity, leading to decreased efficacy. Additionally, octa-substituted T_8_-type silsesquioxanes, often considered candidates for film formation, have limited potential due to their tendency to form three-dimensional structures that reduce mechanical properties. To overcome these challenges, *para*-type (and other adducts) bisfunctional POSS derivatives (Figure 2) have been proposed as an alternative, but the feasibility of their formation is still a matter of ongoing debate.

Given these limitations, our research group has recently reported on functionalized double-decker silsesquioxanes (DDSQs) as a potential alternative to traditional POSS-based materials. The *trans* isomer of DDSQ can serve as a bisfunctional POSS model. We have reported on a family of methacrylate-substituted DDSQs, which exhibit improved mechanical properties and cell adhesion compared to pure organic methacrylates (Figure 6) [51]. To enhance the hydrophilicity of the DDSQ-based composites, polyvinyl alcohol (PVA) was used. The resulting DDSQ-PVA composite was subjected to biological tests, which indicated that human fibroblasts growing on the prepared hybrid composites showed proper spindle-shaped morphology, proliferation, activation status, and increased adhesion and migration abilities similar to those of control conditions. The data obtained from these studies suggest that the prepared DDSQ-based composites can potentially support the wound-healing process. In addition, these materials are relatively easy to prepare, feature simple composition, and are chemically and structurally stable, making them ideal for further development and clinical application.

Double-decker silsesquioxanes can also be used as polarity mediators to obtain hierarchical porous films for bioengineering [52]. Using DDSQs allows for a balance between cell functions, specific surface area, and mechanical properties of the resulting biomaterial. For example, incorporating into the epoxy polymer matrix of the modified glycidyl-propyl DDSQ (G-POSS) made it possible to induce the self-assembly of the polymer chains into three-dimensional porous architecture created by customized apparatus for hierarchical porous film fabrication (Figure 7). Microstructural analysis revealed that incorporating DDSQ particles into the polymer network led to a ca. 30% increase in material strength. Moreover, incorporating silsesquioxanes into the organic matrix affects a decrease in the hydrophobicity of the organic matrix, creating a densely packed 2D or 3D hierarchical porous architecture. Furthermore, the entire composition of the hybrid composite positively affects mouse embryonic fibroblast cells’ behavior toward better adhesion, viability, and proliferation. The resulting hybrid three-dimensional porous film constitutes an attractive environment allowing better nutrition and communication between cells. The obtained composite perfectly mimics the in vivo microenvironment for cells, avoiding their possible morphology changes.

The two examples of applications of DDSQ-based composite in the biomaterials field mentioned above are, according to our best knowledge, the only examples in the literature. In conclusion, the development of functionalized DDSQs represents a significant step forward in wound healing and tissue regeneration, and we look forward to continued advancements in this area.

## 5. Cage-like Silsesquioxanes in Materials Engineering

Because of silsesquioxane cages’ unique advantages, such as environmental neutrality, high thermal stability, fire retardancy, and mechanical performance, it is clear that these materials will find many potential applications in innovative technology and industry. They can find various promising applications, including thermally insulating aerogels, lightweight foams, shielding coatings, thermal management [53], electromagnetic interference shielding [54], protective polymeric films [55], gas barrier films [56], and flame retardancy of polymers [57].

### 5.1. Flame Retardant Nanocomposites

Polymeric materials’ fire retardancy is a significant preoccupation due to the need to minimize fire risk and safety requirements. The combustion process of polymers can be interrupted and retarded by physical processes or chemical reactions of flame retardant (FR) compounds and polymer matrix. The effect of these processes is usually the formation of a protective char layer to prevent the transfer of combustible fuels and oxygen between the flame zone and the pyrolysis zone. Based on structural motive and modes of action, flame retardant compounds are classified as metal hydroxides, halogenated compounds, phosphorus-nitrogen, and silica-containing compounds [58]. From environmental and health points of view, halogenated flame retardants are currently forbidden. On the other hand, silicone-containing FRs improve the flame retardancy of polymers by accumulating silica aggregates in forming a protective layer in the condensed phase.

Silsesquioxanes as nanosized mono/multifunctional POSS cages can be specially designed to increase compatibility and miscibility with polymers and enhance flame retardancy efficiency. In addition, even the low loading level of silsesquioxane nanoparticles in a polymeric matrix such as polycarbonate, polypropylene, and epoxy resin influence reduced flammability, connected with forming an insulating char layer [59,60,61]. However, when using only POSS cages as FR, the residues POSS produced by burning polymers as SiO_2_ aggregates can form porous char on the surface that is next ineffective in the prevention of combustion of the underlying material. Additionally, the industry’s practical application of FR based only on POSS nanoparticles is limited by the high cost of these materials. Therefore, the combination of POSS and conventional FRs is the better option, but the appropriate fitting of components POSS, FR, and polymer matrix is crucial.

The effective PET composite contains two FR components: 9 wt% zinc diethyl hypophosphite and 1 wt% octamethyl-POSS is an example of such a solution [62]. Changing the organic arms attached to the inorganic silsesquioxane core from octamethyl to dodecaphenyl induces the formation of an intumescent char layer with a multicellular structure [63]. The same cages, octamethyl-POSS, and dodecaphenyl-POSS, combined with FR/aluminum phosphinate, were used to prepare flame-retardant PET fibers. In this example, the opposite octamethyl-POSS shows better fire performance than dodecaphenyl-POSS compared to PET fibers without a POSS-FR system [64]. Another example application of silicone cages and FR’s mixture of 3 wt% bisphenyl A-bis (diphenyl phosphate) and 2 wt% trisilanolphenyl-POSS effectively improves the flame retardancy of polycarbonate.

The designing of two-component POSS-FR systems is based on applying classic FR as phosphorous/phosphorous-nitrogen-containing compounds and modification of silicone cages realized by functionalizing the reactive arm of POSS cages, which effectively influences the formed char layer of the compact structure. The comparison of POSS cages with various arms, such as vinyl-POSS, octamethyl-POSS, and dodecaphenyl-POSS, indicated that they form the differential multicellular structure of the char layer [64]. Generally, POSS nanoparticles form effective flame retardants of polymers in the mixture with P/P-N-containing FRs [65,66,67]. The P-containing FR during thermal decomposition gives the free radical quenching effect, and POSS decomposes to SiO_2_, which can further react with phosphate in the char layer [61,68]. The examples of the combination of different P-containing FR and POSS in the polymer matrix are presented in Figure 8. The lower loading level of POSS cages in dual systems was observed in the combination of POSS and P/N-containing intumescent flame retardants (IFRs). The mixture of 7.5 wt% POSS and 22.5% IFR effectively enhances the fire resistance of polylactide. The synergy of the two components is optimal as IFR forms a porous carbonaceous char, but POSS prevents it from thermal degradation and reinforces its mechanical properties [69]. The mixture of POSS and ammonium polyphosphate used on the fire performance of epoxy resins similarly exhibits a synergistic effect, forming an intumescent protective layer [70,71].

The layer-by-layer assembly is the next valuable synthetic strategy employed to prepare flame-retardant polymeric materials by alternatively depositing POSS compound and oppositely charged FRs on the surface of the polymer [72]. The mixture containing 9,10-dihydro-9-oxa-10-phosphaphenanthrene-10-oxide (DOPO) as FR and POSS enhance the mechanical and thermal properties and flame retardancy of polycarbonate (PC) [73,74,75]. As a result, the polymer burns slowly and forms firmer char than PC alone. Similarly, incorporating DOPO-POSS into epoxy resin gives self-extinguishing behavior [76]. Generally, dual systems containing POSS cages and P,N-FRs improve polymers’ flame retardancy and mechanical properties. Although optimal results are desired, structural modification of POSS cages in this area remains inexhaustible. Functionalized silsesquioxane cages with different reactive organic arms are attractive motives in designing synthetic strategies for incorporating FR modules, which allow for obtaining new and effective silsesquioxane-based FR.

The new approaches involve using the composition of several nanomaterials with different geometries, nanoparticles, and nanoplatelets for obtaining polymeric nanocomposites with combined flame resistance and mechanical properties. For example, hybrid FR based on fullerene and graphene has been used in conventional polymers, such as polypropylene [77], phenolic foams [78], and epoxy resin [79]. However, low loading of only 2 wt% of such hybrid fullerene-graphene used as FR in polypropylene matrix indicates the achievement of flame retardancy, but the polymer mechanical properties require improvement [80]. Similarly, as described earlier in nanocomposites, the addition of octaaminophenyl-POSS gives a synergistic effect, enhancing fire safety and mechanical properties [81].

Different types of polymers require a universal strategy for designing FR, and hybrid materials based on combining two or more nanomaterials and conventional FR could reach these requirements. An example of designing such multifunctional FR is material obtained by the reaction of POSS on the surface of graphene and classical DOPO flame retardant applied in a polypropylene matrix. A schematic representation of the combination of multi-structure FR, POSS-Graphene-DOPO (surface functionalization of graphene with POSS and DOPO) applied to PP matrix) is shown in Figure 9.

Graphene and residue of the POSS compound reinforce the char layer and form a physical barrier. Furthermore, phosphorous-containing entities can catalyze the polymer’s carbonization, and during silsesquioxane cages’ degradation, nitrogen residue can form non-flammable volatiles [82]. Although these strategies require more detailed studies, multicomponent nanomaterials with different geometries and conventional flame retardants constitute a prospective strategy for creating advanced FR and safe fire polymers.

Exploring POSS-based hybrid materials as complementary additives, fire retardants, and improved mechanical properties in polymeric materials is still attractive. However, the functionalization of POSS nanocomposites was focused on preparing hybrid materials containing POSS as pendants or end groups [83,84,85,86,87]. Recently, another type of silsesquioxane cages received increasing attention, such as double-decker-shaped silsesquioxanes (DDSQs), which give new possibilities for forming linear hybrid polymers [88]. Synthesis and functionalization of DDSQ cages and polymers with DDSQ in the main chain are well explored, but these materials’ flame retardancy has not been studied so far [89]. However, based on the achievements from the studies of multicomponent POSS-containing and N, P-containing flame retardants in polymeric materials, the change of POSS to DDSQ cages could enhance that effect and increase the safety of commonly used polymers. An example of that solution is the synthesis of the linear copolymer by a hydrosilylation reaction between classical FR phosphorous-containing DOPO and silicon cages DDSQ (Figure 10). DDSQ-DOPO copolymers improve the flame retardancy of PC and ABS composites [90,91].

In polymeric materials, POSS hybrids in multicomponent FR improve thermal and mechanical stability, increase oxidation resistance, and decrease flammability. Furthermore, less explored DDSQs create so-called main chain type polymers or nanocomposites that enhance thermal properties and form new FR for conventional polymeric matrices.

### 5.2. Ziegler-Natta Catalysts

Among many different catalytic systems, heterogeneous Ziegler-Natta catalysts represent the most significant in the polyolefin industry in the context of both extremally high catalytic efficiency and low-cost production [92]. Generally, catalytic systems contain titanium coordination compounds immobilized on MgCl_2_ support activated by trialkyl aluminum. The synthesis of real titanium base catalysts proceeds in situ in the presence of a Lewis base that reacts with titanium precatalysts and magnesium support [93,94,95]. For decades optimization efforts of the industrial process have made it possible to design polyolefin with tailor-made properties [96]. New generations of catalysts enhanced their activity, selectivity, and performance in morphology [97,98,99,100]. However, interactions between components of the catalytic system and especially the formation and distribution of active sites in the presence of appropriate Lewis bases (external or internal electron donors) still require strenuous studies. The simplest is O-donor ligands that, as ethers (THF) and esters (diethylphtalate), can coordinate titanium and/or magnesium centers. Recently, the investigation of the formation of active surface sites in the presence of THF indicates that understanding the role of electron donors is crucial for designing new heterogeneous catalytic systems for olefin polymerization [101]. However, the active site structure is still speculative, and more research is focused not on the catalysts and response to catalyst changes but on its effectiveness and polyolefin yields. The catalysts obtained in situ by sequentially adding titanium compounds, magnesium support, electron internal/external donors, and activator make identifying formed active centers difficult. The DFT calculation and development of in-situ and operando techniques have been developed for decades to study the substantial transformations in all catalyst components during their addition but without spectacular results [102,103,104,105,106]. However, visible progress has been observed in developing these catalysts by modification of MgCl_2_ by unconventional components such as POSS [107]. These studies indicate a promising perspective in effectively improving the distribution of active titanium centers by using this new type of support. Although the Ziegler-Natta type catalysts seem to be exploited entirely, using POSS to form new support for immobilizing titanium precatalysts opens new opportunities for developing Z-N catalysts. Recently, the synthesis of well-defined POSS/MgCl_2_ nano-aggregates and their influence on tuning the coordination behavior of titanium precatalysts were reported. Similarly, the study of new precatalyst MgCl_2_/THF/TiCl_4_ with porous microspheres containing POSS inner cores indicates a new direction for designing heterogeneous catalysts. The presented precatalysts exhibit excellent activity. The porous microspheres with POSS core constitute an attractive model for studying the heterogeneous Ziegler-Natta catalysts for improving their structural parameters and polymer performance. The porous support is an excellent model for the study of the role of Lewis bases in each stage of active site formation during the immobilization of titanium species on MgCl_2_. The presented results indicated that the structural stability of magnesium support is guaranteed by POSS nanocrystals serving as scaffold core for the growth of MgCl_2_ crystals. The obtained polyethylene using POSS/MgCl_2_ support exhibits fine particle size and improved toughness, strength, and stiffness [78].

## 6. Conclusions and Perspectives

In many applications, organic-inorganic composites are the most popular and widely developed hybrid materials. The spectrum of applications is multidimensional, from medicine through everyday materials, to advanced space technologies. Despite the definition of a hybrid, which gives practically infinite possibilities for creating nanocomposites, silicon-based connections are definitely in the lead in the context of the inorganic area. Nanomaterials based on silica or silicone-containing polymers are still attractive. However, in the latest technologies, materials based on silsesquioxane cages of the POSS type and DDSQs have recently occupied a solid and leading position. The simplest method of obtaining hybrid materials based on silsesquioxane cages is to use them as an additive to polymer matrices to modify their properties, often involving a new function or expanding the application potential of the original material. Material chemistry often focuses on cages’ properties using their excellent thermomechanical parameters and biocompatibility. Hence, there are many publications about new materials, which are actually “old” ones only with the addition of POSS cages with functional arms that allow for further modifications. These materials will become widely available over time as long as the commercial synthesis of silsesquioxane cages is cost-effective and industrially scalable. This is one of the most anticipated trends in transferring scientific research to industrial implementation. However, the research and design of innovative materials with new functions require not reaching for commercially available additives (POSS, DDSQ, etc.); it is crucial to combine the efforts of synthetics offering unique cage systems and to incorporate them into the structures of other materials, such as an advancement in material design with tailor-made function, literally made-to-order [108,109,110]. One question remains whether material chemistry is interesting in this field because, currently, cages are used mainly instrumentally, considering only their classical possibilities, i.e., add if the goal is to increase thermal stability or improve mechanical properties. Cages offer much more, and that is required is a less conventional approach.

## Data Availability

Not applicable.

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
