# Peer review of "A Brief Review on Selected Applications of Hybrid Materials Based on Functionalized Cage-like Silsesquioxanes"

_polymers, 2023, doi:10.3390/polym15061452_

Round 1

Reviewer 1 Report

Please provide a maximum of 5 keywords.

Please rewrite the abstract in order to find the significance of this review article.

The introduction part is very short.

Author Response

Dear Reviewer

Thank you for your kind and valuable evaluation of our manuscript. Please find below our replies to your remarks/comments. All changes we made are highlighted in red in the revised manuscript.

With due respect

Lukasz John

Query 1: "Please provide a maximum of 5 keywords."

Reply: The number of keywords is reduced to 5.

Query 2: "Please rewrite the abstract in order to find the significance of this review article."

Reply: The abstract has been rewritten to emphasize the significance of the review paper.

Query 3: "The introduction part is very short."

Reply: The authors agree that the introduction is short, but we think it is adequate for the content of the review paper. We have decided to leave this as it is.

Reviewer 2 Report

This manuscript mainly summarizes the research on the potential application fields of functional caged silsesquioxane hybrid materials in recent years, mainly polyhedral oligomeric silsesquioxane (POSS) and double-decker silsesquioxane (DDSQs). In this manuscript, the author mainly introduces the potential applications of functional caged silsesquioxane hybrid materials from the perspective of biomaterials, flame retardants and catalytic materials. In general, the author has summarized many new research reports and conclusions, but there are still many problems to be solved. Therefore, I think this paper can only be reconsidered after fully addressing several major issues as noted below:

1. The characteristics of POSs and DDSQ with clear definition and thermal stability are emphasized in the abstract, but their application in the field of biomaterials is given priority in the process of describing their application. It is suggested to sort out and add the features emphasized in the abstract so as to correspond with the following text logically. A similar problem is that the relationship between structure, physical chemistry and mechanical properties is proposed in the abstract, but it is not enough in the content of application introduced later.

2. This article focuses on summarizing the frontier progress of POSS and DDSQ in the application field, but it takes a lot of space to explain the particularity of the entire silicon-based species from the field of basic chemistry, which can reduce the length of this part of discussion and help readers understand the two core structures of POSS and DDSQ faster.

3. The subtitles 4.1 and 4.2 of the introduction of biomaterials are changed to "polyhedral oligomeric silsesquioxane..." instead of "caged silsesquioxane...", which may be more consistent with the discussion.

4. Comprehensive conclusions should be supplemented with corresponding references, such as lines 398-399.

5. Adjust and replace the pictures to make the style of the full text pictures consistent, such as Figure 9 and Figure 10.

6. Subtitle 5.1.1 is unnecessary, and it may be more logical to change 5.1.2 to 5.2.

7. Although the author's research focus is on the application of POSS and DDSQ in biomaterials, as a summary article of application direction, the application in materials engineering is still insufficient and can be supplemented appropriately.

8. The authors could add the following references which would again increase the interest to general functional caged silsesquioxane readers: Chinese Journal of Polymer Science, 2020, 38, 1149-1156; Nanoscale, 2020, 12, 11395-11415; ACS Macro Letters‚ 2021‚ 10, 1563-1569.

Author Response

Dear Reviewer

Thank you for your kind and valuable evaluation of our manuscript. Please find below our replies to your remarks/comments. All changes we made are highlighted in red in the revised manuscript.

With due respect

Lukasz John

Query 1: "The characteristics of POSs and DDSQ with clear definition and thermal stability are emphasized in the abstract, but their application in the field of biomaterials is given priority in the process of describing their application. It is suggested to sort out and add the features emphasized in the abstract so as to correspond with the following text logically. A similar problem is that the relationship between structure, physical chemistry and mechanical properties is proposed in the abstract, but it is not enough in the content of application introduced later."

Reply: The abstract has been written to emphasize the content of the review paper.

An issue referring to the relationship of the structure and physical properties based on attractive thermal properties of the cage-like systems, both for POSS and DDSQ, are nontrivial in the chemistry of silicon-based compounds versus materials engineering. Unfortunatelly, in the literature, cage-like silsesquioxanes are mainly added as agents improving, for instance, the mechanical properties of the entire material. From this point of view, in our opinion, there is no need to analyze such properties in detail. This is also not described in referenced papers. The specific structure that possesses both kinds of caged silsesquioxanes derives from the properties described in paragraph 2, entitled "What is unique about silicon-based compounds?". We think that this paragraph describing the uniqueness of such structure lies in the specific bond energy (Si-O), specific reactivity of Si-H compared to the C-H bond, and thermal superiority of three coexisting siloxanes Si-O moieties forming the cage. In the described papers, DDSQ or POSS are mainly added as "dopants" rather than agents modifying, e.g., polymer matrix, etc., which are covalently bonded to other components of the entire system.

Query 2: "This article focuses on summarizing the frontier progress of POSS and DDSQ in the application field, but it takes a lot of space to explain the particularity of the entire silicon-based species from the field of basic chemistry, which can reduce the length of this part of discussion and help readers understand the two core structures of POSS and DDSQ faster."

Reply: The specific structure that possesses both kinds of caged silsesquioxanes derives from the properties described in paragraph 2, entitled "What is unique about silicon-based compounds?". This paragraph describing the uniqueness of such structure lies in the specific bond energy (Si-O), specific reactivity of Si-H compared to the C-H bond, and thermal superiority of three coexisting siloxanes Si-O moieties forming the cage. This paragraph is necessary to understand cage-like silsesquioxanes' architecture and stability (thermal or chemical). In the literature reports, DDSQs or POSSs are mainly added as "dopants" rather than agents modifying, e.g., polymer matrix, etc., which are covalently bonded to other components of the entire system. In turn, paragraph 3 strictly defines both kinds of cage-like silsesquioxanes. We think that this description should be straightforward.

Query 3: "The subtitles 4.1 and 4.2 of the introduction of biomaterials are changed to "polyhedral oligomeric silsesquioxane..." instead of "caged silsesquioxane...", which may be more consistent with the discussion."

Reply: POSS or DDSQ species possess various synonyms, for instance: cage-like structures, cage-like silsesquioxanes, cubic silsesquioxanes, etc. The authors of the paper used these synonyms alternately.

Query 4: "Comprehensive conclusions should be supplemented with corresponding references, such as lines 398-399."

Reply: All adequate references are placed in appropriate places.

Query 5: "Adjust and replace the pictures to make the style of the full text pictures consistent, such as Figure 9 and Figure 10."

Reply: The requested Figures have been changed.

Query 6: "Subtitle 5.1.1 is unnecessary, and it may be more logical to change 5.1.2 to 5.2."

Reply: The authors agree with this remark. All necessary changes have been made.

Query 7: "Although the author's research focus is on the application of POSS and DDSQ in biomaterials, as a summary article of application direction, the application in materials engineering is still insufficient and can be supplemented appropriately."

Reply: In a brief review, the authors wanted to focus on selected applications and some chemical aspects of reported systems. Adding more information in the field of materials engineering will unnecessarily increase the volume of the manuscript and will be detrimental to the main purpose of such a review paper.

Query 8: "The authors could add the following references which would again increase the interest to general functional caged silsesquioxane readers: Chinese Journal of Polymer Science, 2020, 38, 1149-1156; Nanoscale, 2020, 12, 11395-11415; ACS Macro Letters‚ 2021‚ 10, 1563-1569."

Reply: Mentioned-above references have been added to the manuscript.

Round 2

Reviewer 2 Report

I think the current revised version seems OK for me.